# Stage-Dependent Activity and Pro-Chondrogenic Function of PI3K/AKT during Cartilage Neogenesis from Mesenchymal Stromal Cells

**DOI:** 10.3390/cells11192965

**Published:** 2022-09-23

**Authors:** Felicia A. M. Klampfleuthner, Benedict Lotz, Tobias Renkawitz, Wiltrud Richter, Solvig Diederichs

**Affiliations:** 1Research Centre for Experimental Orthopaedics, Department of Orthopaedics, Heidelberg University Hospital, D-69118 Heidelberg, Germany; 2Department of Orthopaedics, Heidelberg University Hospital, D-69118 Heidelberg, Germany

**Keywords:** mesenchymal stromal cells, chondrogenesis, cartilage, chondrocyte hypertrophy, PI3K, AKT, PKB, LY294002, insulin

## Abstract

Differentiating mesenchymal stromal cells (MSCs) into articular chondrocytes (ACs) for application in clinical cartilage regeneration requires a profound understanding of signaling pathways regulating stem cell chondrogenesis and hypertrophic degeneration. Classifying endochondral signals into drivers of chondrogenic speed versus hypertrophy, we here focused on insulin/insulin-like growth factor 1 (IGF1)-induced phosphoinositide 3-kinase (PI3K)/AKT signaling. Aware of its proliferative function during early but not late MSC chondrogenesis, we aimed to unravel the late pro-chondrogenic versus pro-hypertrophic PI3K/AKT role. PI3K/AKT activity in human MSC and AC chondrogenic 3D cultures was assessed via Western blot detection of phosphorylated AKT. The effects of PI3K inhibition with LY294002 on chondrogenesis and hypertrophy were assessed via histology, qPCR, the quantification of proteoglycans, and alkaline phosphatase activity. Being repressed by ACs, PI3K/AKT activity transiently rose in differentiating MSCs independent of TGFβ or endogenous BMP/WNT activity and climaxed around day 21. PI3K/AKT inhibition from day 21 on equally reduced chondrocyte and hypertrophy markers. Proving important for TGFβ-induced SMAD2 phosphorylation and SOX9 accumulation, PI3K/AKT activity was here identified as a required stage-dependent driver of chondrogenic speed but not of hypertrophy. Thus, future attempts to improve MSC chondrogenesis will depend on the adequate stimulation and upregulation of PI3K/AKT activity to generate high-quality cartilage from human MSCs.

## 1. Introduction

Large focal cartilage injuries in the knee are routinely treated with autologous articular chondrocytes (ACs), although their harvest creates novel non-regenerating cartilage defects. Mesenchymal stromal cells (MSCs) are a promising alternative since they are readily available from regenerative tissues like bone marrow and can differentiate into chondrocytes that deposit cartilage tissue rich in proteoglycans and type II collagen [1,2]. However, instead of developing an AC-like phenotype, MSCs undergo endochondral development in vitro and phenocopy hypertrophic growth plate chondrocytes with characteristic markers like type X collagen, Indian hedgehog (IHH), parathyroid hormone 1 receptor (PTH1R), alkaline phosphatase (ALP), and integrin-binding bone sialoprotein (IBSP), and with the capacity to induce ectopic bone formation [1,3,4]. Importantly, ACs under identical culture conditions generate permanent cartilage that is not remodeled into bone in vivo [4].

Analyzing the deviant responses of MSCs and ACs to identical culture conditions has revealed signaling pathways that drive MSCs into the endochondral lineage. Stimulated with transforming growth factor beta (TGFβ), MSCs reduce TGFβ receptor expression, upregulate bone morphogenetic proteins (BMPs) and IHH, and maintain a high cell-autonomous WNT activity while ACs show an opposite regulation. TGFβ receptors, BMP antagonists, and the natural IHH inhibitor PTHrP (parathyroid hormone-related protein) are upregulated, and WNT activity is silenced [3,5,6,7]. Since they upregulate both chondrocyte and hypertrophy markers [1,2,7,8,9], we consider TGFβ and BMPs to be pro-chondrogenic regulators of the speed of MSC chondrogenesis, while WNT and IHH/PTHrP are regulators of hypertrophy that leave chondrocyte markers largely unaffected [3,8,10]. In this context, the role of one of the most prominent stimulators of chondrocyte hypertrophy in the growth plate, the insulin-like growth factor I (IGF1)-triggered phosphoinositide 3-kinase (PI3K)/AKT pathway [11] remained so far uninvestigated.

Chondrogenic MSC and AC cultures are standardly treated with serum-free medium containing high doses of insulin (5–10 µg/mL) that activate PI3K/AKT signaling upon binding to its cognate insulin receptor IR (reviewed by [12]) but can also activate the IGF1 receptor, IGF1R [13,14]. However, whether this continuous stimulation of PI3K/AKT via IR and IGF1R mainly regulates the speed of MSC chondrogenesis by equally stimulating chondrocyte and hypertrophy markers or whether PI3K/AKT signaling drives chondrocyte hypertrophy remains so far unaddressed.

Importantly, in the growth plate, chondrocyte markers and cartilage matrix deposition were largely independent of PI3K/AKT signaling according to a variety of knockout models. Neither a deficiency in IGF1, the receptors IGF1R or IR, nor a single or double knockout of AKT1 and AKT2 delayed or reduced chondrocyte markers, including proteoglycan deposition in the zones of resting and proliferating chondrocytes [15,16,17,18,19]. Moreover, IGF1 infusion did not increase chondrocyte matrix production in the growth plates of hypophysectomized rats [20]. Cultured chondrocytes and cartilage tissue, by contrast, appeared more dependent on PI3K/AKT signaling, and treatment with insulin or IGF1 is important for their matrix production and chondrocyte marker expression [21,22,23,24,25,26]. Overexpression and inhibitor studies showed that PI3K/AKT signaling stimulated proteoglycan and type II collagen deposition along with *ACAN* and *SOX9* expression in human AC alginate and monolayer cultures, bovine ACs, and rat nucleus pulposus cells [27,28,29,30]. Altogether, these studies suggested that the importance of PI3K/AKT signaling for chondrocyte markers and cartilage matrix production may be context-dependent.

In the context of MSC in vitro chondrogenesis, we have previously shown that PI3K/AKT activity was important for cell proliferation only during the first two weeks but not later on [31], and cell proliferation was in turn required for proper chondrogenesis and type II collagen deposition [32]. Whether in the later phase of MSC chondrogenesis, PI3K/AKT signaling may drive chondrocyte hypertrophy remains so far uninvestigated.

A major pro-hypertrophic activity of PI3K/AKT signaling was demonstrated by a large variety of developmental animal studies. The knockout of IGF1 or IGF1R, or the combined knockout of AKT1 and AKT2 severely affected the zone of hypertrophic growth plate chondrocytes which was reduced in length and contained less hypertrophic (i.e., smaller in size) chondrocytes [16,17,33,34], which ultimately reduced bone length, delayed ossification, and impaired skeletal mineralization [15,17,19,35,36,37]. For knockouts of the insulin receptor, similar but less severe effects were reported [18,38]. Conversely, IGF1 or insulin stimulation enlarged the hypertrophic zone and the hypertrophic chondrocytes in embryonic mouse or rat metatarsal explants, and juvenile rabbit tibiae [39,40,41,42] and increased the expression of *Ihh*, *Col10a1*, and other hypertrophic markers in monolayer cultures of chicken growth plate chondrocytes and a chondrogenic rat cell line [43,44,45]. Given such strong evidence, it appears important to consider a potential pro-hypertrophic role of PI3K/AKT signaling also during MSC chondrogenesis in vitro.

Interestingly, in osteogenic human MSC cultures, ALP enzyme activity, osteopontin, and osteocalcin protein levels, matrix mineralization and other characteristics that osteoblasts share with hypertrophic chondrocytes depended on PI3K/AKT activity [46,47,48,49]. Thus, although a strong body of literature demonstrated that PI3K/AKT signaling is strongly associated with chondrocyte hypertrophy and pro-osteogenic cues, its exact contribution to the chondrogenesis and hypertrophic development of human MSCs remains largely unclear.

The aim of this study was therefore to unravel the role of PI3K/AKT signaling for MSC chondrogenesis with a special focus on its putative pro-chondrogenic versus pro-hypertrophic function during the later phase of chondrogenesis. The activation of PI3K/AKT signaling in chondrogenic 3D MSC pellet cultures was compared with ACs, and its regulation by signaling pathways that drive either the speed of chondrogenesis (TGFβ, BMP) or hypertrophy (WNT) was assessed. To categorize the function of PI3K/AKT signaling, the dose–response of chondrocyte markers, proteoglycan deposition, and hypertrophy markers under PI3K inhibition was investigated. Better knowledge of the importance and function of PI3K/AKT signaling for MSC chondrogenesis and hypertrophy may improve our ability to guide MSCs between chondral versus endochondral pathways and improve cell-based cartilage and bone regeneration strategies.

## 2. Materials and Methods

### 2.1. Isolation and Expansion of MSCs and ACs

All procedures were approved by the local Ethics Committee on human experimentation of the Medical Faculty of Heidelberg University and in accordance with the Helsinki Declaration of 1975 in its latest version. MSCs were isolated from human bone marrow aspirates obtained with informed consent from patients (age 21–83 years) undergoing total hip replacement. Mononuclear cells were isolated via Ficoll-Paque^TM^ (Cytiva, Freiburg, Germany) as described previously [2] and expanded for three passages in medium composed of high glucose-containing Dulbecco’s Modified Eagle’s Medium (DMEM; Gibco, Life Technologies, Thermo Fisher Scientific, Carlsbad, CA, USA), 12.5% fetal calf serum (FCS; Gibco, Life Technologies, Thermo Fisher Scientific, Carlsbad, CA, USA), 1% penicillin/streptomycin (Biochrom, Bioswisstec, Schaffhausen, Switzerland), 2 mM L-glutamine, 1% non-essential amino acids, 1% β-mercaptoethanol (all Gibco, Life Technologies, Thermo Fisher Scientific, Carlsbad, CA, USA), and 4 ng/mL human fibroblast growth factor-2 (FGF-2, Active Bioscience, Hamburg, Germany). Cells were cultured at 37 °C and 6% CO_2_, and the medium was exchanged three times a week.

ACs were isolated from human articular cartilage obtained with informed consent from patients (age 49–77) undergoing total knee replacement. Cartilage from phenotypically healthy regions was minced and digested overnight at 37 °C with 1.5 mg/mL collagenase B (Roche Diagnostics, Basel, Switzerland) and 0.1 mg/mL hyaluronidase (Sigma-Aldrich, St. Louis, MO, USA). Chondrocytes were expanded for two passages with medium comprising low-glucose DMEM supplemented with 10% FCS and 1% penicillin/streptomycin.

### 2.2. Cartilage Neogenesis from MSCs and ACs

MSCs and ACs were cultured as 3D pellets (5 × 10^5^ cells/pellet) in chondrogenic medium consisting of high-glucose DMEM, 0.1 µM dexamethasone, 0.17 mM ascorbic acid-2 phosphate, 4 mM sodium pyruvate, 0.35 mM proline, 5 µg/mL transferrin, 5 ng/mL selenous acid, 1.25 mg/mL bovine serum albumin (all from Sigma-Aldrich, St. Louis, MO, USA), 1% penicillin/streptomycin, 1% ITS+ premix (Corning Life Sciences, New York City, NY, USA) or similar amounts of insulin (Lantus, Sanofi-Aventis, Frankfurt, Germany), and 10 ng/mL recombinant human TGFβ1 (Biomol, Hamburg, Germany) for up to 6 weeks with medium changes three times a week.

For the indicated time points during MSC chondrogenesis, the chondrogenic medium was supplemented with the BMP inhibitor LDN-212854 (LDN-21; 500 nM in DMSO solvent, day 0–42; Sigma-Aldrich, St. Louis, MO, USA), the WNT-inhibitor IWP-2 (2 μM in DMSO solvent, day 14–42), or LY294002 (LY; 0.25 µM–25 µM in DMSO, day 21–42; both from Tocris Bioscience, Bristol, United Kingdom). When appropriate, the controls were treated with DMSO. Where indicated, TGFβ1 was withdrawn from day 21 onward. For the Western blot detection of phosphorylated AKT, pellets were harvested at the designated days 48 h after the last medium exchange.

### 2.3. Quantitative Gene Expression Analysis

Total RNA was isolated by phenol/guanidine isothiocyanate extraction using peqGOLD Trifast (PEQLAB, Erlangen, Germany) according to the manufacturer’s instructions. Complementary DNA was synthesized from 500 ng of total RNA using Omniscript R reverse transcriptase (Qiagen, Hilden, Germany). Transcript levels were determined by quantitative PCR (qPCR) analysis using SybrGreen (Thermo Fisher Scientific, Waltham, MA, USA) and the LightCycler R 96 system (Roche Diagnostics, Basel, Switzerland) with the gene-specific primers shown in Appendix A. Gene expression was normalized to the Ct value of the reference genes *RPL13* and *CPSF6*, and the relative gene expression was calculated as 100% × 1.8^−ΔCt^.

### 2.4. Western Blotting

Two pellets per time point and group were pooled in 120 µL PhosphoSafe Extraction Reagent (Merck Millipore, Darmstadt, Germany) supplemented with 1 mM Pefabloc^®^ SC (Sigma-Aldrich, St. Louis, MO, USA) and minced in a mixer mill (Retsch, Haan, Germany) for 2 × 2 min at 30 Hz. After centrifugation for 20 min at 13,000× *g* and 4 °C, the supernatants were transferred to clean reaction tubes, mixed with Laemmli buffer (33.2% (*w*/*v*) glycerine (Carl Roth, Karlsruhe, Germany), 249mM Tris-HCl pH 6.8, 8.0% (*w*/*v*) SDS, 0.02% bromphenol blue (all from Sigma-Aldrich, St. Louis, MO, USA) in distilled water), and heated for 5 min at 95 °C. The samples were separated by denaturing sodium-dodecyl sulfate polyacrylamide gel electrophoresis and blotted on nitrocellulose (Amersham^TM^, GE Healthcare, Chalfont St Giles, United Kingdom). To probe for several proteins of interest in a sample, the membrane was cut horizontally at 50 kDa. The following antibodies were used: rabbit polyclonal anti-pAKT antibody (Ser473, 1:500, #9271), rabbit polyclonal anti-AKT antibody (1:1000, #9272), rabbit monoclonal anti-pSMAD2 antibody (Ser465/467, 1:500, #3108, clone 138D4), monoclonal anti-SMAD2/3 antibody (1:1000, #8685, clone DFG7), monoclonal anti-pSMAD1/5/9 antibody (Ser463/465, Ser463/465, Ser465/467, 1:250, #13820, all from Cell Signaling Technologies, Danvers, MA, USA), monoclonal anti-SMAD1/5 (SMAD1: 1:500, ab33902; SMAD5: 1:1000, ab40771; both from Abcam, Berlin, Germany), rabbit polyclonal anti-SOX9 antibody (1:2000, AB5535, Merck Millipore, Darmstadt, Germany), or mouse monoclonal anti-β-actin antibody (1:10,000, GTX26276, clone AC-15, GeneTex, Irvine, CA, USA). Bands were detected by peroxidase-coupled goat anti-mouse antibody (1:5000, #115-035-071) or peroxidase-coupled goat anti-rabbit antibody (1:10,000, #111-035-046, both from Jackson ImmunoResearch Laboratories, West Grove, PA, USA) and visualized by enhanced chemiluminescence (Roche Diagnostics, Basel, Switzerland).

### 2.5. ALP Enzyme Activity

Culture supernatants from four to five replicate pellets per group and time point were pooled, and 100 µL was incubated with 100 µL of substrate solution [10 mg/mL p-nitrophenyl phosphate (Sigma-Aldrich, St. Louis, MO, USA) in 0.1 M Glycin (Carl Roth, Karlsruhe, Germany), 1 mM MgCl_2_, and 1 mM ZnCl_2_ (both from Sigma-Aldrich, St. Louis, MO, USA), pH 9.6]. After 120 min, the absorbance was recorded at 405/490 nm (FLUOStar OMEGA, BMG LABTECH, Ortenberg, Germany). The substrate conversion was referred to a p-nitrophenol-derived standard curve (Sigma-Aldrich, St. Louis, MO, USA) and calculated as ALP enzyme activity (ng/mL/min).

### 2.6. GAG and DNA Quantification

For the measurement of the glycosaminoglycan (GAG) and DNA content, two pellets per group were pooled and digested with 3 U/mL proteinase K (Thermo Fisher Scientific, Waltham, MA, USA) dissolved in 0.05 M Tris (Merck Millipore, Darmstadt, Germany) and 1 mM CaCl_2_ (pH 8.0, Sigma-Aldrich, St. Louis, MO, USA) overnight at 60 °C. The DNA content of the digest was determined with the Quant iT PicoGreen ds DNA Assay Kit (Thermo Fisher Scientific, Waltham, MA, USA). For GAG quantification, 30 μL of the 1:2 diluted proteinase K digest was mixed with 200 μL of 1,9-dimethyl-methylene blue (DMMB, Sigma-Aldrich, St. Louis, MO, USA) dye solution (pH 3.0) [50], the absorbance was measured at 530 nm, and it was referred to a chondroitin sulfate standard curve.

### 2.7. Histology

The pellets were fixed in 4% formaldehyde for 2 h, dehydrated, and paraffin-embedded. Sections of 5 µm thickness were deparaffinized, rehydrated, stained with Safranin O (0.2% in 1% acetic acid; Fluka, Sigma-Aldrich, St. Louis, MO, USA), and counterstained with Fast Green (0.04% in 0.2% acetic acid; Merck Millipore, Darmstadt, Germany) according to standard histological protocols. Immunohistological staining of type II collagen was performed as described previously [2]. In brief, sections were treated with 4 mg/mL hyaluronidase (Sigma-Aldrich, St. Louis, MO, USA) followed by 1 mg/mL pronase (Roche Diagnostics, Basel, Switzerland), and unspecific binding sites were blocked with 5% bovine serum albumin. Type II collagen was detected with a monoclonal mouse anti-human type II collagen antibody (1:1000, 0863171-CF, clone II-4C11, MP Biomedicals, Santa Ana, CA, USA) and visualized with biotinylated goat anti-mouse secondary antibody (1:500, Dianova, BIOZOL, Hamburg, Germany), streptavidin-alkaline phosphatase (Dako, Agilent, Santa Clara, CA, USA), and Fast Red (Vector Laboratories, Newark, CA, USA).

### 2.8. Statistical Analyses

The number of independent experiments performed for each analysis is given in the figure captions. For all values, the mean and the standard error of the mean were calculated, and differences between groups were analyzed using the Mann–Whitney *U* test (MWU) or LSD-corrected ANOVA for multiple comparisons, as specified in the figure captions. A two-tailed significance value of *p* ≤ 0.05 was considered statistically significant. The data were analyzed using SPSS-25 (IBM, Armonk, NY, USA).

## 3. Results

### 3.1. Inverse Regulation of PI3K/AKT Activation in MSC versus AC Chondrogenic Cultures

To document the chondrogenic capacity of MSCs and the phenotypic differences between hypertrophic MSC-derived chondrocytes and ACs, we compared matrix deposition and specific marker expression of human MSC versus AC 3D pellet cultures in standard serum-free chondrogenic medium containing 10ng/mL TGFβ1 and 6.25 µg/mL insulin. Histological analyses revealed a similar accumulation of cartilage extracellular matrix rich in proteoglycans and type II collagen in both groups after 6 weeks and a round cell morphology that is characteristic of chondrocytes (Figure 1A,B). The ACs started with significantly higher expression of the chondrocyte marker *COL2A1* but differentiating MSCs reached *COL2A1* levels of ACs at days 21 and 42 of differentiation (Figure 1C). Of note, MSC-derived chondrocytes expressed *COL10A1* and *IBSP* at significantly higher levels than AC samples on days 21 and 42 (Figure 1D) along with significantly higher ALP enzyme activity from day 21 on (Figure 1E). Taken together, these data confirmed the successful differentiation of MSCs and ACs with the important difference that MSCs but not ACs followed the endochondral route and developed into hypertrophic chondrocytes.

Next, we investigated PI3K/AKT activity over the course of the chondrogenic differentiation of MSCs in comparison to ACs. The protein levels of phosphorylated AKT (pAKT) were compared at weekly intervals via Western blotting using β-actin as a reference. Interestingly, the pAKT levels rose during the first three weeks of chondrogenesis in all seven investigated MSC donor samples and peaked in a donor-dependent manner between days 14 and 28 (Figure 1F,G), which was independent of donor age. In one out of the seven donor samples, the pAKT levels were already high at day 7. In six out of the seven MSC donor samples, the pAKT levels then declined until day 42. In strong contrast, AKT phosphorylation in ACs declined donor-dependently from day 7 or day 14 of re-differentiation on and remained at low levels further on (Figure 1F,G). Importantly, a direct comparison of the pAKT/β-actin ratio in MSC and AC samples on the same membrane demonstrated significantly higher AKT phosphorylation in MSC cultures throughout days 21 to 42, although the pAKT levels declined again in MSC-derived chondrocytes (Figure 1H). Taken together, an inverse PI3K/AKT regulation was evident during endochondral MSC development versus chondral AC re-differentiation and showed a characteristic high level in the middle phase of MSC chondrogenesis.

### 3.2. Elevated AKT Activation Is Independent of TGFβ and Cell-Autonomous BMP and WNT Activity

The PI3K/AKT pathway is a central signaling hub that can be influenced by many other signaling pathways, and we next asked whether signaling pathways that are known to deviate between chondrogenic MSC and AC cultures could be responsible for elevated pAKT levels in MSC pellets. Specifically, we considered pro-chondrogenic TGFβ and BMP activity as well as pro-hypertrophic WNT signaling and tested whether their reduction would decrease AKT phosphorylation in MSC pellets. Discontinuing the TGFβ1 treatment of MSC cultures in the phase of high AKT activation from day 21 (no TGFβ in Figure 2A–C) still allowed high proteoglycan and type II collagen deposition, according to the histological assessment of the day 42 samples in line with previous data [51], albeit some reduction in proteoglycan and type II collagen deposition was apparent (Figure 2A). Surprisingly, the pAKT protein levels were elevated in samples from four out of five independent MSC donor populations compared to controls at all investigated time points after TGFβ1 withdrawal (Figure 2B,C). Samples from a fifth donor showed no consistent response. This indicated that TGFβ reduced AKT phosphorylation in MSC-derived chondrocytes.

We next inhibited endogenous BMP activity with 500 nM LDN-212854 (LDN-21), a concentration that was capable to block BMP-induced SMAD1/5/9 phosphorylation and that significantly reduced ALP activity at day 42 (Appendix A), whereas proteoglycan and type II collagen deposition during MSC chondrogenesis was largely maintained (Figure 2D). Importantly, the upregulation of the pAKT protein levels remained unchanged, indicating that AKT activity was independent of endogenous BMP signaling (Figure 2E,F). WNT inhibition with 2 µM IWP-2 starting at day 14 of MSC chondrogenesis, as previously established [3,52], strongly reduced ALP activity at maintained proteoglycan and type II collagen deposition (Figure 2G, Appendix A) but did also not affect AKT phosphorylation (Figure 2H,I). Overall, none of the investigated pro-chondrogenic or pro-hypertrophic signaling pathways that discriminate chondrogenic MSC from AC cultures appeared to be directly responsible for the increased AKT activation in the MSC cultures, thus leaving the classification of AKT signaling in one or the other pathway category undetermined.

### 3.3. Elevated PI3K/AKT Activation Is Essential for High Proteoglycan Deposition

To elucidate whether PI3K/AKT signaling is a pro-chondrogenic or a pro-hypertrophic pathway during MSC chondrogenesis, we inhibited PI3K with 25 µM LY294002 (LY) starting at day 21. According to our previous data, cell survival and proliferation were not affected by this LY dose between days 14 and 28 of MSC differentiation [31] Control pellets were treated with the respective amount of the solvent DMSO. Western blotting confirmed a strong reduction in AKT phosphorylation under LY treatment compared to controls at all investigated time points (Figure 3A, Appendix A). Of note, 25 µM LY reduced the gene expression of the hypertrophy marker *COL10A1* by 81% compared to the control samples at day 42, and *IHH* and *PTH1R* were significantly lowered by 95% and 76%, respectively (Figure 3B, Appendix A). Additionally, 25 µM LY significantly reduced the osteogenic marker *IBSP* mRNA by 94% compared to the control and fully suppressed ALP activity in culture supernatants from day 28 on (Figure 3C,D). However, this strong anti-hypertrophic effect of 25 µM LY came at the expense of a diminished proteoglycan and type II collagen content in the pellets (Figure 3E, Appendix A). Proteoglycan quantification revealed a 70% reduction by 25 µM LY compared to control pellets (Figure 3F). The histologically apparent smaller pellet size was reflected by a reduced cell content according to DNA quantification (Figure 3G) after three weeks of LY treatment. Importantly, also the expression of the chondrogenic markers *COL2A1* and *ACAN* was suppressed in the day 42 samples (Figure 3H). Altogether, this demonstrated that PI3K suppression was strongly anti-chondrogenic rather than specifically anti-hypertrophic and that PI3K/AKT activity was essential in the late phase of MSC chondrogenesis to maintain chondrocyte marker expression and high cartilage matrix production. Since ACs were capable to maintain high cartilage matrix production despite low AKT activation, we next asked whether reducing instead of strongly suppressing pAKT levels in MSC-derived chondrocytes would allow to maintain cartilage matrix production and decrease hypertrophic markers.

Briefly, 0.25–2.5 µM LY dose-dependently reduced AKT phosphorylation in differentiating MSCs at all investigated time points according to Western blotting (Figure 4A, Appendix A). The expression of the hypertrophy markers *COL10A1*, *IHH*, and *PTH1R* remained unchanged with 0.25 µM and was slightly but not significantly reduced by 1.25 µM and 2.5 µM LY at day 42 (Figure 4B, Appendix A). Osteogenic markers *IBSP*, *ALPL*, and ALP enzyme activity in culture supernatants remained, however, unaffected by all of these LY concentrations (Figure 4C,D, Appendix A). Importantly, 0.25–2.5 µM LY allowed homogeneous proteoglycan and type II collagen deposition according to the histological assessment of the day 42 samples (Figure 4E, Appendix A). However, proteoglycan staining appeared increasingly weaker than in solvent controls, and the GAG/DNA levels were significantly reduced by 1.25 µM and 2.5 µM LY (−19% and −27%, resp., Figure 4F) at unchanged DNA content (Figure 4G). Like hypertrophy markers, *COL2A1* and *ACAN* mRNAs were also dose-dependently reduced (2.5 µM: −50% *COL2A1*, −46% *ACAN*, Figure 4H), albeit this remained a trend. Thus, the reduction in chondrocyte hypertrophy by decreasing PI3K/AKT activation came only at the expense of chondrogenic power. This indicated that the observed elevated PI3K/AKT activation in late MSC chondrogenesis was an essential pro-chondrogenic driver of chondrocyte markers and cartilage matrix deposition.

### 3.4. PI3K/AKT Activity Is Important for TGFβ Signaling and High SOX9 Protein Levels

To further substantiate the pro-chondrogenic PI3K/AKT function and elucidate a potential underlying mechanism, we next assessed the importance of PI3K/AKT for TGFβ/SMAD2 signaling and the chondrogenic master transcription factor SOX9, which are both crucial for chondrocyte marker expression and cartilage matrix synthesis. Western blotting revealed a strong reduction in pSMAD2 levels under treatment with 25 µM LY compared to the controls (Figure 5A, Appendix A) and a dose-dependent pSMAD2 reduction by 0.25–2.5 µM LY (Figure 5B, Appendix A). Thus, TGFβ-induced SMAD2 signaling was impaired by AKT inhibition, indicating that high PI3K/AKT activity was necessary for proper TGFβ signaling in differentiating MSC cultures.

In addition, 25 µM LY strongly reduced SOX9 protein levels in differentiating MSC cultures at all tested time points (Figure 5C, Appendix A), and a SOX9 reduction was also observed with 2.5 µM but not with lower LY doses (Figure 5D, Appendix A). Conclusively, our data demonstrated that elevated PI3K/AKT activity in MSC-derived chondrocytes was crucial for two pro-chondrogenic factors, the SMAD2-dependent canonical TGFβ activity and the chondrogenic master transcription factor SOX9, and thus for their downstream targets, including *COL2A1* gene expression and proteoglycan production.

## 4. Discussion

To re-direct MSC chondrogenesis from the intrinsic endochondral into the desired chondral lineage that gives rise to phenotypically stable AC-like chondrocytes and articular cartilage neogenesis requires a profound knowledge of the signaling pathways that regulate chondrocyte differentiation and drive hypertrophy. Surprisingly, the contribution of the PI3K/AKT pathway to these two processes inherent to MSC in vitro chondrogenesis has not been addressed yet, even though the IGF1/PI3K/AKT axis is important for cartilage matrix deposition by chondrocytes and is a prominent driver of chondrocyte hypertrophy in the growth plate. We here showed for the first time that an inverse regulation of PI3K/AKT activation discriminated MSC chondrogenesis from AC re-differentiation in vitro, which demonstrated a chondrocyte lineage or differentiation stage-dependent response of chondrogenic cultures to PI3K/AKT stimulation. Interestingly, rising PI3K/AKT activation in differentiating MSCs appeared independent of TGFβ treatment and endogenous BMP or WNT activity, and was thus not driven by these lineage-dependent signals with known differential regulation in chondrogenic MSC and AC cultures. Importantly, our data demonstrated for the first time, that in the later phase of MSC chondrogenesis, PI3K/AKT equally drives chondrocyte and hypertrophy markers and is important for activating the pro-chondrogenic mediator SMAD2, as well as for the protein accumulation of the transcription factor SOX9, a master enhancer of cartilage matrix gene expression. Thus, our data demonstrated that PI3K/AKT signaling is a pro-chondrogenic rather than a pro-hypertrophic pathway in TGFβ-stimulated chondrogenic cultures.

A very interesting and novel discovery of our study was the so far undetected inverse regulation of PI3K/AKT stimulation by MSCs and ACs under identical chondrogenic culture conditions. Apparently, upregulated PI3K/AKT activity was not specific for endochondral differentiation because on the one hand, major endochondral signaling pathways did not upregulate AKT phosphorylation, and on the other hand, PI3K/AKT was not pro-hypertrophic. Chronologically, the phase of high PI3K/AKT activity appeared to coincide with cells in the stage of proliferating chondroblasts, which, according to our previous data, is reached between days 10 and 21 of MSC differentiation, when the cells re-entered the cell cycle and started to deposit type II collagen [31,32]. This indicated that the regulation of PI3K/AKT signaling was differentiation stage-dependent rather than chondrocyte lineage-dependent. This interpretation is consistent with developmental mouse studies that reported immune histochemical pAKT detection in proliferating and pre-hypertrophic chondrocytes and strongly diminished levels in the hypertrophic chondrocyte zone of the growth plate [53,54]. Moreover, this assumption was also consistent with previous observations in the ATDC5 cell line, where the insulin-induced recruitment of PI3K to the membrane and downstream AKT phosphorylation were high at the proliferative stage but low at the differentiated, hypertrophic chondrocyte stage [42]. Based on a reduced IR abundance upon insulin treatment, the authors suggested a ligand-induced receptor downregulation as a potential mechanism. While this mechanism was in line with the here observed repression of PI3K/AKT activity in ACs, it was not in line with the transiently upregulated PI3K/AKT response observed during MSC chondrogenesis. We therefore suggest a differentiation stage-dependent mechanism with a high PI3K/AKT response characteristic for proliferating chondroblasts.

Of note, not only PI3K/AKT activity but also its function appeared to be differentiation stage-dependent. In a previous study, we showed that PI3K/AKT activity was initially required for cell proliferation during the early phase (first 14 days) of MSC chondrogenesis [31]. Our current data now showed that PI3K/AKT activity remained essential throughout later MSC differentiation but with a pro-chondrogenic instead of a proliferative function. Overall, PI3K/AKT signaling functions as a required stimulator of the speed of chondrogenesis in vitro. This pro-chondrogenic role of PI3K/AKT was not expected from the IGF1/IGF1R/IR/PI3K/AKT knockout animal models, where authors described growth plate defects but did not report proteoglycan amounts to be affected [15,16,17,18,19,33,34,35,55]. However, in the complex in vivo system, the knockout of a single gene can easily be compensated by alternative receptors, ligands, or effector isoforms. Interestingly, when the insulin receptor was knocked out in cultured mouse chondrocytes in vitro, *SOX9* expression and proteoglycan deposition were reduced compared to the controls, even though in the same study, a cartilage matrix phenotype in IR-deficient chondrocytes in the growth plate was not apparent [38]. Additionally, in vitro, LY treatment reduced alcian blue-stained proteoglycans along with the *Col2a1* expression of mesenchymal mouse limb bud cells cultured in serum-containing medium [56]. Together with our results and the well-documented regulation of PI3K/AKT downstream of either IGF1 or insulin for proteoglycan deposition in chondrocyte cultures, these studies suggest that the pro-chondrogenic effects of PI3K/AKT deficiency can be compensated in the growth plate.

We here further substantiated the pro-chondrogenic PI3K/AKT function by the discovery of its requirement for TGFβ-induced SMAD2 phosphorylation and for the accumulation of the SOX9 protein. Crosstalk between PI3K/AKT and the TGFβ signaling pathways has been reported in various different cells and tissues, and both stimulatory and inhibitory effects were reported depending on the cell context [57,58]. While ours is the first report of an AKT-dependent SMAD2 activation in TGFβ-stimulated chondrocytes, further mechanistic studies are needed to illuminate whether this was mediated by supporting SMAD2 localization to the TGFβ receptor as suggested for a mouse epithelial cell line and HeLa cells [59,60,61]. A stimulation of SOX9 gene expression and protein accumulation by PI3K/AKT signaling similar to our observations has previously been reported for human ACs, human fetal growth plate chondrocytes, and nucleus pulposus cells [29,30,62]. Importantly, Yin et al. demonstrated that constitutively active AKT was capable to stimulate *SOX9* and *COL2A1* expression in human ACs in the absence of TGFβ treatment [30], thus indicating pro-chondrogenic AKT function beyond being a mere mediator of TGFβ signaling.

Although a pro-hypertrophic role of PI3K/AKT is well-established in the growth plate, and despite all parallels between MSC and growth plate chondrocyte differentiation, our data demonstrated that PI3K/AKT did not contribute to hypertrophy during MSC in vitro chondrogenesis. This was convincingly substantiated by three main findings in our study. First, the pro-hypertrophic signaling network that comprises WNT upstream of IHH and BMP activity [3] was not directly responsible for the elevated AKT response during MSC chondrogenesis. Secondly, PI3K/AKT inhibition reduced the hypertrophic phenotype solely at the expense of chondrogenic power. Thirdly, AKT strongly contributed to TGFβ/SMAD2 signaling and SOX9 accumulation, which are all pro-chondrogenic factors. Thus, PI3K/AKT inhibition was no means to uncouple cartilage neogenesis from the hypertrophic degeneration of MSC-derived chondrocytes. In line, while IGF1 was previously reported to upregulate hypertrophy markers in MSC-derived chondrocytes, chondrocyte markers were always increased simultaneously [63,64], which further supported a pro-chondrogenic rather than a pro-hypertrophic role for PI3K/AKT in MSC cultures. With this in mind, it may appear surprising that for MSCs under osteogenic instead of chondrogenic stimulation, PI3K/AKT was reported to stimulate ALP levels, mineralizing activity, osteopontin, and osteocalcin protein levels, which are all characteristics that osteoblasts share with hypertrophic chondrocytes [46,47,48,49]. A main difference between the osteogenic and chondrogenic differentiation of MSCs is the application of TGFβ, whose activity is enhanced by PI3K/AKT according to our data. Without TGFβ, PI3K/AKT could well be expected to affect differentiating MSCs differently. In contrast to our culture model, TGFβ signaling in the hypertrophic chondrocyte zone of the growth plate is low according to diminished TGFβR1 protein levels along with the reduced activation of SMAD2/3 [65]. This invites the speculation that the pro-hypertrophic activity of PI3K/AKT could become obvious in the absence of external TGFβ. Since the stimulation of chondrocyte hypertrophy was undesired here and in the greater context of cartilage regeneration, we did not test this hypothesis by inhibiting AKT during the stimulation of MSC-derived chondrocytes with TGFβ-free hypertrophic medium. Taken together, we believe that the PI3K/AKT effects in the later phase of MSC chondrogenesis are attributed to stimulating known pro-chondrogenic factors, which explains its importance for cartilage matrix production and the virtual absence of pro-hypertrophic activity.

One limitation of our study was that ACs were isolated from cartilage samples of osteoarthritic joints. Although we only harvested cartilage tissue that was macroscopically intact, we cannot exclude that the tissue source affected the chondrocyte phenotype. As a quality control, however, we demonstrated by careful characterization that re-differentiated ACs did not exhibit a hypertrophic phenotype. Therefore, we are convinced that the here used ACs represented fully adequate populations of non-hypertrophic chondrocytes.

Another important point to consider is our use of an artificial inhibitor for studying the importance of PI3K/AKT instead of reducing or discontinuing insulin treatment. However, potential compensation by cell-autonomous IGF1 secretion and possible further influences by numerous pathways, including TGFβ, as shown here, could result in the underestimation of the PI3K/AKT importance. Of course, disregarded off-target effects could potentially compromise the validity of attributing an LY effect to the sole reduction in PI3K activity. However, for LY, the described off-targets are either AKT downstream targets (mTOR1, GSK3β) [66,67], which we anyway expected to affect, or regulators of cell cycle progression (CK2, PLK1, PIM1, PIM3) [67,68], which would be inconsistent with our previous observation that LY was not mitogenic in the late stage of MSC chondrogenesis [31].

In conclusion, we here described for the first time that the insulin-stimulated PI3K/AKT activation in chondrogenic cultures was differentiation-stage dependent and upregulated in proliferating chondroblasts but repressed in differentiated chondrocytes. Having previously shown that PI3K/AKT activity stimulates cell proliferation during early MSC chondrogenesis, we here revealed that it was later required for SMAD2 phosphorylation and SOX9 accumulation and thus a pro-chondrogenic stimulator of cartilage matrix deposition. Our data identified PI3K/AKT signaling in the presence of TGFβ as a required stimulator of the speed of chondrogenesis and not a regulator of hypertrophy like in the growth plate. This new knowledge underscores how strongly cartilage neogenesis in vitro can depend on a highly potent insulin formulation. Future attempts to improve cartilage neogenesis from MSCs in vitro, to overcome the undesired hypertrophic cartilage degeneration, and to bring MSCs closer to clinical application for cartilage regeneration, will depend on an adequate stimulation and upregulation of PI3K/AKT activity to generate high-quality cartilage from human MSCs.

## Figures and Tables

**Figure 1 cells-11-02965-f001:**
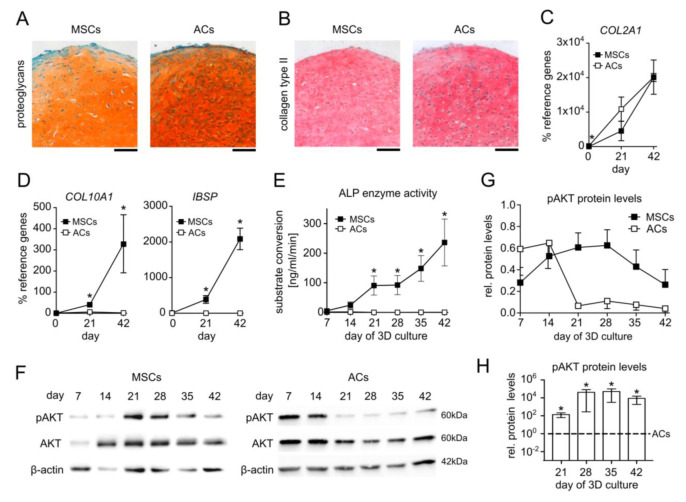
Inverse AKT regulation in hypertrophic MSC versus AC chondrogenic cultures. Pellets of expanded MSCs and ACs were subjected to chondrogenic culture for 6 weeks in the presence of TGFβ1 and insulin. Samples were harvested at the indicated time points 48 h after medium exchange. (**A**,**B**) Paraffin sections of MSC and AC pellets at day 42 stained with Safranin O/Fast Green to visualize proteoglycan deposition and for type II collagen by immunohistochemistry. Scale bars: 100 µm. MSCs: n = 8, ACs: n = 5. (**C**,**D**) Gene expression analysis of the chondrogenic marker *COL2A1*, the hypertrophic marker *COL10A1*, and the osteogenic marker *IBSP* at days 0, 21, and 42 of MSC chondrogenesis and AC re-differentiation. *RPL13* and *CPSF6* were used as reference genes; n = 3. (**E**) Alkaline phosphatase (ALP) activity at weekly intervals was determined in culture supernatants pooled from four MSC or AC pellets for each independent experiment. MSCs: n = 6–7, ACs: n = 4. (**F**) Western blot analysis of pAKT and AKT protein using β-actin as loading control. One representative of n = 6–7 for MSCs from six independent donors and n = 4 for ACs is shown. (**G**) Densitometric analysis of Western blots shown in (**F**). (**H**) Densitometric analysis of Western blots: pAKT/β-actin protein levels in MSCs and ACs at indicated time points were compared on the same membrane; n = 3. Given n-numbers refer to the number of experiments performed with independent MSC or AC donor populations. Means ± SEM are shown. * *p* ≤ 0.05 MSCs vs. ACs, Mann–Whitney *U* test (MWU).

**Figure 2 cells-11-02965-f002:**
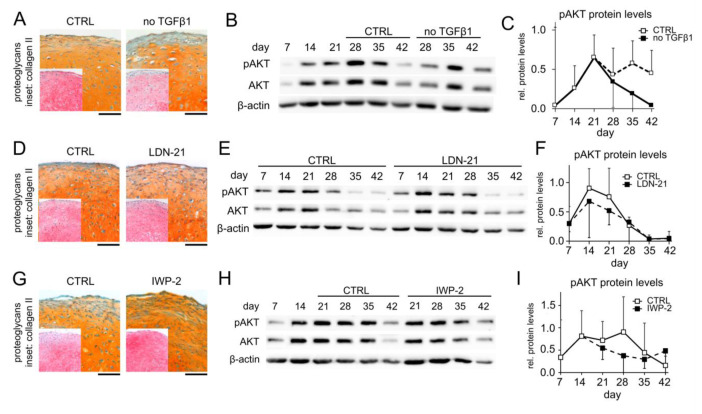
Elevated AKT activation is independent of TGFβ and cell-autonomous BMP and WNT activity. MSC pellets were subjected to chondrogenic culture for 6 weeks and the effects of TGFβ, BMP, and WNT signaling on AKT activation were investigated. (**A**–**C**) TGFβ1 treatment was discontinued from day 21 on in the treatment groups (no TGFβ1), whereas control pellets received 10 ng/mL TGFβ1 throughout 6 weeks of pellet culture (CTRL). (**D**–**F**) 500 nM of the BMP inhibitor LDN-212854 (LDN-21) or 0.02% DMSO solvent (CTRL) was applied from day 0 on. (**G**–**I**) MSC pellets were treated with 2 µM IWP-2 or 0.04% DMSO solvent (CTRL) from day 14 on. (**A**,**D**,**G**) Paraffin sections of day 42 MSC pellets were stained with Safranin O/Fast Green to visualize proteoglycan deposition and by immunohistochemistry to visualize type II collagen (insets). Scale bars: 100 µm for proteoglycans and 200 µm for type II collagen. (**B**,**C**,**E**,**F**,**H**,**I**) Western blot detection of pAKT and AKT protein levels using β-actin as loading control. Medians ± 95% confidence intervals are shown; n = 3–5.

**Figure 3 cells-11-02965-f003:**
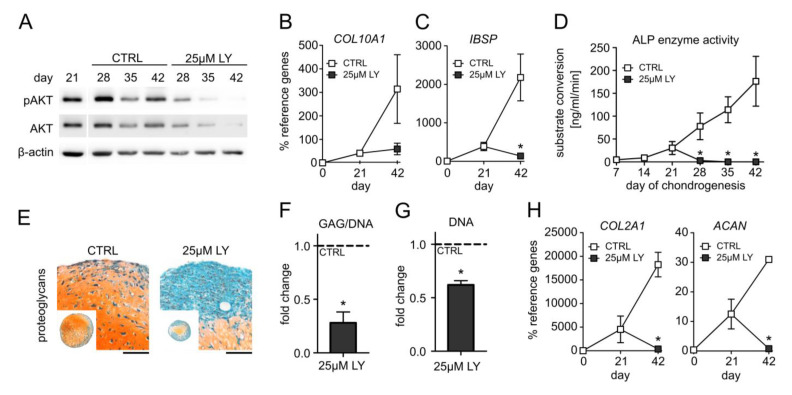
AKT suppression impairs chondrogenesis of MSCs. MSC pellets were subjected to chondrogenic culture for 6 weeks and treated with 25 µM LY294002 (LY) or 0.1% DMSO solvent (CTRL) starting from day 21 of chondrogenesis. Samples were harvested at indicated time points. (**A**) Western blot analysis for pAKT and AKT using β-actin as loading control. Shown is one representative of three independent experiments. All samples were loaded on the same gel and detected on the same membrane. (**B**,**C**) Gene expression analysis by qPCR for *COL10A1* and *IBSP* at days 0, 21, and 42 using *CPSF6* and *RPL13* as reference genes. (**D**) Alkaline phosphatase (ALP) activity was determined in culture supernatants pooled from four pellets per group and time point. (**E**) Paraffin sections of MSC pellets at day 42 were stained for proteoglycans with Safranin O/Fast Green or for type II collagen by immunohistochemistry. Scale bars: 100 µm for enlarged sections and 1 mm for overviews. Shown is one representative of three independent experiments. (**F**) GAG content at day 42 was assessed by DMMB assay and referred to DNA quantity. (**G**) DNA amount per pellet at day 42 was quantified by Pico Green assay. (**H**) Gene expression analysis of *COL2A1* and *ACAN* by qPCR at days 0, 21, and 42 using *CPSF6* and *RPL13* as reference genes. Experiments were performed in three independent MSC donor populations (n = 3). Graphs show means ± SEM. * *p* ≤ 0.05 compared to CTRL at the same time point, MWU.

**Figure 4 cells-11-02965-f004:**
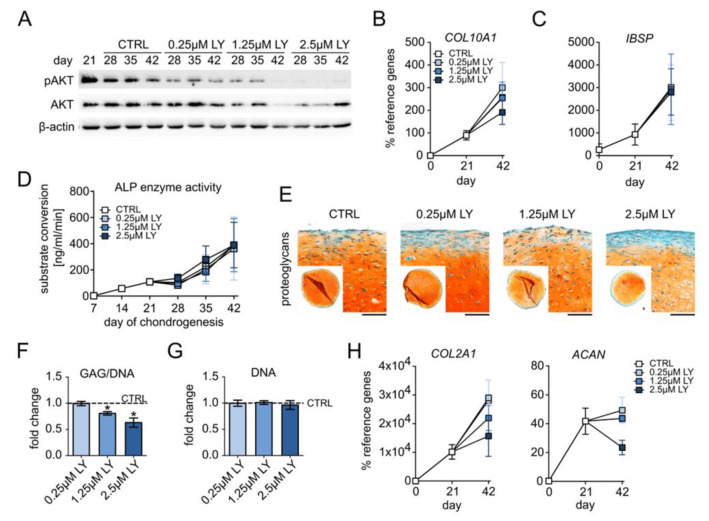
Pro-chondrogenic instead of pro-hypertrophic function of PI3K/AKT activity. MSC pellets were subjected to chondrogenic culture for 6 weeks and treated with 0.25 µM, 1.25 µM, or 2.5 µM LY294002 (LY) or 0.01% DMSO solvent (CTRL), starting from day 21 of chondrogenesis. (**A**) Western blot analysis for pAKT and AKT protein levels using β-actin as loading control. Shown is one representative of three independent experiments. (**B**,**C**) Gene expression analysis by qPCR for the hypertrophy marker *COL10A1* and the osteogenic marker *IBSP* using *CPSF6* and *RPL13* as reference genes. (**D**) ALP enzyme activity was determined in pooled supernatants from four pellets per group and time point. (**E**) Paraffin sections of day 42 MSC pellets were stained with Safranin O/Fast Green to visualize proteoglycan deposition. Scale bars: 100 µm for enlarged sections and 1mm for overviews. (**F**) GAG content at day 42 was assessed via DMMB assay and referred to the DNA amount. (**G**) DNA content per pellet at day 42 was quantified via Pico Green assay. (**H**) Gene expression analysis by qPCR for the chondrogenic markers *COL2A1* and *ACAN* using *CPSF6* and *RPL13* as reference genes. n = 3 independent MSC donor populations for each experiment. Graphs show means ± SEM. * *p* ≤ 0.05.

**Figure 5 cells-11-02965-f005:**
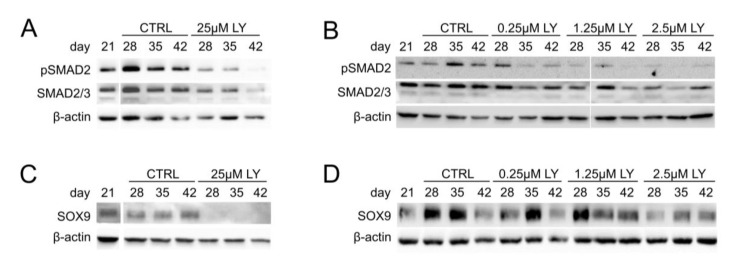
PI3K/AKT signaling is important for TGFβ activity and SOX9 protein accumulation. MSC pellets were subjected to chondrogenic culture for 6 weeks and treated with the indicated concentration of LY or DMSO (0.1% for A, C; 0.01% for B, D) from day 21 of chondrogenesis on. Western blot analyses for pSMAD2 and total SMAD2/3 (**A**,**B**) as well as SOX9 protein (**C**,**D**) using β-actin as loading control. All samples were loaded on the same gel and detected on the same membrane. Shown is one representative of three independent experiments.

## Data Availability

The data presented in this study are available on reasonable request from the corresponding author.

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
