# Peer review of "Stage-Dependent Activity and Pro-Chondrogenic Function of PI3K/AKT during Cartilage Neogenesis from Mesenchymal Stromal Cells"

_cells, 2022, doi:10.3390/cells11192965_

Round 1
Reviewer 1 Report
In this manuscript, the authors concluded that PI3k/Akt signaling drives chondrocyte differentiation, and that this pathway is independent of TGFB/BMP/WNT pathways. The authors also concluded that differentiation stage-dependent mechanism with high PI3K/Akt response by proliferating chondrocytes. The role of PI3k/Akt in mediating cell proliferation is well known and as such, remains unclear what new values these findings will add to the field. Further comments are as follows:
1) PI3K/AKT pathway is also a major regulator of survival. A key concern is that in many of the experiments, a concentration of 25uM LY294002 was used (e.g. Figure 3, Figure 5A, C). This concentration caused reduction in pAKT, AKT and B-actin, and DNA contents, clearly suggesting induction of cell toxicity/death as this concentration was too high. While lower concentrations (0.25-2.5uM) did appear to inhibit pro-chondrogenic pathway without affecting cell survival, it isn't clear why the authors used 25uM LY294002 and this requires detailed explanation in the manuscript.
2) There are some inconsistencies in pAKT activations in different blots. For instance, Figure 1F, pAKT increased and peaked at Day 21/28 for MSCs. Figure 2B, pAKT was high in D7 but decreased overtime for CTRL MSCs. Could the authors explain this inconsistency?
3) Densitometry analysis is required for all the western blot figures.
4) Day 0 images and blots were not included in the figures as controls, and thus need to be included.
Author Response
Thank you very much for your valuable comments. All changes in the manuscript are highlighted.
Reviewer #1:
“In this manuscript, the authors concluded that PI3k/Akt signaling drives chondrocyte differentiation, and that this pathway is independent of TGFB/BMP/WNT pathways. The authors also concluded that differentiation stage-dependent mechanism with high PI3K/Akt response by proliferating chondrocytes. The role of PI3k/Akt in mediating cell proliferation is well known and as such, remains unclear what new values these findings will add to the field.”
Response: We agree that PI3K/AKT is a known regulator of cell proliferation, and as we describe in detail in our introduction (lines 77-82), we have previously shown that PI3K/AKT activity was important for cell proliferation during MSC chondrogenesis, but, importantly, only during the first two weeks. Between days 14 and 28, PI3K/AKT activity did not regulate cell proliferation. In the current study we now show, that PI3K/AKT is still required during the later phase of MSC chondrogenesis, because it functions as a pro-chondrogenic regulator of the speed of chondrogenesis. We rephrased our conclusion in lines 542-543 to make this clear: “Having previously shown that PI3K/AKT activity stimulates cell proliferation during early MSC chondrogenesis, we here revealed that PI3K/AKT it was later required for SMAD2 phosphorylation and SOX9 accumulation, and thus a pro-chondrogenic stimulator of cartilage matrix deposition.”
1) PI3K/AKT pathway is also a major regulator of survival. A key concern is that in many of the experiments, a concentration of 25uM LY294002 was used (e.g. Figure 3, Figure 5A, C). This concentration caused reduction in pAKT, AKT and B-actin, and DNA contents, clearly suggesting induction of cell toxicity/death as this concentration was too high. While lower concentrations (0.25-2.5uM) did appear to inhibit pro-chondrogenic pathway without affecting cell survival, it isn't clear why the authors used 25uM LY294002 and this requires detailed explanation in the manuscript.
Response: We here followed up a previous study, where treatment with 25µM LY did not affect cell proliferation during days 14 and 28 of MSC chondrogenesis (Fischer et al. 2018). In fact, this LY dose has frequently been used to treat chondrocytes without affecting cell survival according to DNA quantification (Starkman et al. Biochem J 2005; Liu-Bryan et al. J Immunol 2005; Xue et al. Biomed Pharmacother 2013; He et al. Cell Physiol Biochem 2016). We clarified this in the results section (lines 315-317): “According to our previous data, cell survival and proliferation were not affected by this LY dose between days 14 and 28 of MSC differentiation (Fischer et al. 2018).”
2) There are some inconsistencies in pAKT activations in different blots. For instance, Figure 1F, pAKT increased and peaked at Day 21/28 for MSCs. Figure 2B, pAKT was high in D7 but decreased overtime for CTRL MSCs. Could the authors explain this inconsistency?
Response: MSC chondrogenesis is well known to be highly variable between donor populations. The here observed variable time point of maximal PI3K/AKT activity appeared to coincide with the stage of proliferating chondroblasts and is, thus, expected to vary between donor populations. We thank the reviewer for spotting that in Figure 2B we did not choose a typical representative with a clear pAKT upregulation after day 7. We carefully checked all our raw data again and found in our total of 9 independent donor populations only 1 with a high pAKT level already on day 7. We now chose a more representative example for Figure 2B, but keep the data of the less typical experiments in our study, since there was no obvious reason to exclude them. Variability is reflected in the densitometric evaluations of our Western blots, with the less typical samples included in the quantifications in Figure 1G, where we now included all standard controls (n=7 from 6 independent donors, DMSO-treated controls are not included in this figure), and in figure 2C, where the variability on day 7 is very high due to the one less typical experiment. That pAKT is consistently upregulated but the time course of this upregulation is variable can now clearly be seen in the line diagrams of the densitometric Western blot analyses. We changed the results text as follows (lines 237-242): “Interestingly, pAKT levels rose during the first three weeks of chondrogenesis in all 7 investigated MSC donor samples and peaked in a donor-dependent manner between day 14 and 28 (Figure 1F, G), which was independent of donor age. In 1 out of the 7 donor samples, pAKT levels were already high at day 7.”
3) Densitometry analysis is required for all the western blot figures.
Response: We added densitometry to all Western blot figures (Figures 1, 2, Supplemental Figures 2, 3, 4).
4) Day 0 images and blots were not included in the figures as controls, and thus need to be included.
Response: We indeed included d0 initially in our analyses, which is why they are marked on the original blots that were included in our submission. However, we had to exclude our d0 data from all analyses for the following reasons: On day 0, cells are not pelleted but grow in monolayer with FCS-containing expansion medium instead of chondrogenic medium. Cell harvest on day 0 was performed with trypsin, while pellets were snap-frozen with liquid nitrogen. The time between trypsinization of monolayer cells and transfer to -80°C was not standardized. Thus, we cannot directly compare Western blot bands of a phosphorylated signalling protein like pAKT that we detected in d0 samples with those at all other time points.
Reviewer 2 Report
I congratulate the authors for presenting an interesting study.
Some studies have reported that MSCs from aged mice express higher levels of AKT as compared to MSCs from young mice. Others have also reported that ex vivo passaging of MSCs increases the expression of stress kinases in them. Since in your study, you have taken BM aspirate samples from patients aged 21-83 years, I am quite speculative that you have got consistently inverse correlation of PI3K/AKT activation in MSCs versus AC chondrogenic cultures. Could you provide a clarification of the same?
Author Response
Thank you very much for your valuable comments. All changes in the manuscript are highlighted.
Reviewer #2:
Some studies have reported that MSCs from aged mice express higher levels of AKT as compared to MSCs from young mice. Others have also reported that ex vivo passaging of MSCs increases the expression of stress kinases in them. Since in your study, you have taken BM aspirate samples from patients aged 21-83 years, I am quite speculative that you have got consistently inverse correlation of PI3K/AKT activation in MSCs versus AC chondrogenic cultures. Could you provide a clarification of the same?
Response: We carefully checked our data and found no correlation between maximal pAKT levels and donor age (please find an overview of pAKT peaks and donor age in the attachment). We accordingly changed our text in the results section (lines 237-241): “Interestingly, pAKT levels rose during the first three weeks of chondrogenesis in all 7 investigated MSC donor samples and peaked in a donor-dependent manner between day 14 and 28 (Figure 1F, G), which was independent of donor age.”

Round 2
Reviewer 1 Report
I believe the authors have sufficiently addressed my concerns and revised accordingly. Therefore, I think the manuscript is now acceptable for publication.